# Establishing the Prevalence of Osteomalacia in Arab Adolescents Using Biochemical Markers of Bone Health

**DOI:** 10.3390/nu14245354

**Published:** 2022-12-16

**Authors:** Nasser M. Al-Daghri, Sobhy Yakout, Shaun Sabico, Kaiser Wani, Syed Danish Hussain, Naji Aljohani, Suma Uday, Wolfgang Högler

**Affiliations:** 1Chair for Biomarkers of Chronic Diseases, Biochemistry Department, College of Science, King Saud University, Riyadh 11451, Saudi Arabia; 2Obesity Endocrine and Metabolism Center, King Fahad Medical City, Riyadh 12231, Saudi Arabia; 3Department of Endocrinology and Diabetes, Birmingham Women’s and Children’s Hospital, Steelhouse Lane, Birmingham B4 6NH, UK; 4Institute of Metabolism and Systems Research, University of Birmingham, Edgbaston, Birmingham B15 2TT, UK; 5Department of Pediatrics and Adolescent Medicine, Johannes Kepler University Linz, 4020 Linz, Austria

**Keywords:** biochemical osteomalacia, nutritional rickets, bone markers, vitamin D deficiency, children

## Abstract

Nutrition-acquired osteomalacia is a bone mineralization disorder associated with dietary calcium and/or solar vitamin D deficiency, risk factors considered common in the Middle Eastern region. Establishing less invasive, cheap, and widely available diagnostic markers for this underdiagnosed entity is essential, in particular for screening in high-risk groups. This study assessed the prevalence of biochemical osteomalacia in Arab adolescents. In this cross-sectional study performed between September 2019 and March 2021, adolescents aged 12–17 years from 60 different secondary and preparatory year schools in Riyadh, Saudi Arabia were included. Anthropometrics and fasting blood samples were collected. Biochemical osteomalacia was defined as any two of the following four serum markers of hypomineralization, namely low 25 hydroxyvitamin D (25OHD < 30 nmol/L), high alkaline phosphatase (ALP), low calcium (Ca), and/or inorganic phosphorous (Pi). A total of 2938 Arab adolescents [1697 girls; mean age (years) 14.8 ± 1.8; 1241 boys; mean age 15.1 ± 1.6] were recruited. Vitamin D deficiency was noted in 56.2% (n = 953) of girls and 27.1% (n = 336) of boys (*p* < 0.001). The overall prevalence of biochemical osteomalacia was 10.0% (n = 295/2938) and was higher in girls than boys (14.7% vs. 3.6%, *p* < 0.001). The prevalence of low serum Ca and/or Pi was also higher in girls than in boys (24.2% vs. 12.5%, respectively, *p* < 0.001), as well as elevated ALP (5.1% vs. 1.5%, *p* < 0.001). Overall, girls were 4.6 times (95% CI 3.3–6.4) more likely to have biochemical osteomalacia than boys. Screening of apparently healthy Arab adolescents revealed a high prevalence of deranged mineralization markers suggestive of biochemical osteomalacia, which was significantly more common in girls than boys and was likely associated with Arab traditional clothing and diet. The proposed combination of typically altered mineralization markers for the diagnosis of osteomalacia is, at best, suggestive until further comparisons with established diagnostic tools (histological analysis of bone biopsies) are conducted.

## 1. Introduction

The global prevalence of vitamin D deficiency and insufficiency (defined as serum 25 hydroxyvitamin D [25(OH)D] < 30 and 30–50 nmol/L) ranges from ~5–18% to 24–49%, respectively, depending on the study region and population [1]. Insufficient dietary vitamin D intake (below the estimated average requirement [EAR] of 10 μg/day) exists within populations around the globe, further exacerbating the vitamin D deficiency pandemic [1]. This is not surprising given that dietary sources of vitamin D are very few unless food is fortified with vitamin D and, therefore, humans rely on sunlight to synthesize vitamin D. In Saudi Arabia, the overall prevalence of low vitamin D (defined as 25OHD < 50nmol/L) in adults was 63.5%, based on a recent meta-analysis involving 16 epidemiological studies [n = 20,787 individuals] [2]. However, this prevalence jumps to 81% when other population groups, such as newborns, children, and adolescents, as well as pregnant and lactating women are included [3]. Given the pivotal role of vitamin D in bone and calcium metabolism, aggressive public health campaigns, including several local and regional guidelines, were released to address this high prevalence of vitamin D deficiency within the Arabic community [4,5]. While these strategies have led to modest success in terms of improving the vitamin D status of the community over time [6], several populations, children and adolescents, in particular, remain vulnerable to the complications of vitamin D deficiency, including nutritional rickets and osteomalacia, disorders that are likely underdiagnosed and relatively understudied in the Middle East region.

Osteomalacia is a metabolic and systemic skeletal disorder in children and adults with impaired mineralization of pre-formed bone matrix, alongside malaise, pain, and muscle weakness [7]. In young children, the bone disorder osteomalacia manifests in conjunction with the growth plate disorder rickets and frequent hypocalcaemic symptoms. In adolescents, the clinical presentation is that of osteomalacia growth plates are fusing [8]. By far the most common cause of nutritional rickets and osteomalacia in children is the combination of solar vitamin D with dietary calcium deficiencies [9]. These micronutrient deficiencies, if left untreated, predispose children to complications such as myopathy, delayed development, bony deformities, fractures, seizures, tetany and, in infants, hypocalcaemic dilated cardiomyopathy, to name a few [9]. Given the high level of underdiagnosis, it has been proposed that a prevalence of nutritional rickets in children higher than 1% should warrant a public health response [10]. However, population-based studies on nutritional rickets remain limited and even more so for osteomalacia since it requires an invasive bone biopsy for confirmation.

Studies on nutritional rickets and osteomalacia in Saudi Arabia are limited overall. A small (n = 66) single-center study of children aged <3 years conducted between 1997 and 1999 estimated the incidence of rickets at 0.5%, based on clinical symptoms, biochemistry, and observations of wrist and knee radiographs [11]. Another single-center retrospective study (1990–2009) study in the central region reported 81 children (2–18 years) with confirmed rickets (58%) and osteomalacia, based on clinical, biochemical, and radiologic features, [12]. Bone pain was the most commonly reported clinical presentation of osteomalacia, defined by serum alkaline phosphatase (ALP) ≥500 IU/L in a retrospective review conducted between 2000 and 2006 in the central region [22 (38%) out of 57 children aged 10–16 years] [13]. Finally, in a 2016 study involving Saudi adolescent girls, the prevalence of clinical [bone pain and muscle weakness] and biochemical features suggestive of osteomalacia [25(OH)D < 25 nmol/L, elevated ALP, normal or low calcium (Ca), and inorganic phosphorous (Pi)] was 2.1% (33 out of 1538) [14]. One limitation of that study was the exclusion of boys and the biases inherent in the reporting of pain and weakness [14].

Taking into consideration the role of dietary calcium intake, Uday and Högler recently proposed a non-invasive approach focusing on nutritional osteomalacia, which requires the presence of high ALP, high parathyroid hormone (PTH), low dietary Ca intake (<300 mg/day), and/or low serum 25(OH)D (<30 nmol/L) [8]. Given the equivocal definition of biochemical osteomalacia, the present cross-sectional study aims to determine its presence among Arab adolescents using four of the well-known, easily and inexpensively analyzable serum markers of mineralization, namely serum 25(OH)D, ALP, Ca, and Pi.

## 2. Materials and Methods

In this cross-sectional study, healthy Saudi Arabian students, boys and girls, aged 12–17 years at the time of recruitment, were invited from around 60 secondary and preparatory year schools in Riyadh, Saudi Arabia, from September 2019 until March 2021, in cooperation with the Ministry of Education. Adolescents outside the eligible age range, with acute medical conditions, those who were not fit to consent, non-Saudis, and non-ambulatory participants were excluded. Anthropometrics were assessed, and fasting blood and serum samples were collected for biochemical analysis relevant to bone mineralization and micronutrient deficiencies. All blood samples were stored at the Biobank of the Chair for Biomarkers of Chronic Diseases (CBCD), the College of Science at King Saud University, Riyadh, Saudi Arabia. Ethical approval was obtained from the Institutional Review Board (IRB) of the College of Medicine, King Saud University (E-21–6095, approved 18 January 2019; amended 7 April 2022).

### 2.1. Anthropometry

Height (cm) and weight (kg), were measured by certified nurses assigned to each school in each consenting participant during all visits, as performed in previous school studies [15,16]. The certified nurses were trained to participate in epidemiological and screening studies and were, therefore, well-versed in collecting data and biological samples for large-scale research studies. Height was measured in centimeters (cm) using a standardized stadiometer. Weight was obtained to the nearest 0.1 kg using a Detecto balance beam scale (Detecto Scale Inc, Brooklyn, NY, USA). Body Mass Index (BMI) was calculated as weight in kg divided by height in squared meters (kg/m^2^).

### 2.2. Blood Samples

Venous blood samples were collected in the morning following an overnight fast of 8–10 h. Samples were centrifuged on-site, the serum was separated, placed on ice, and immediately transported to the laboratory of the CBCD at King Saud University, where blood and serum were stored at −80 °C. Serum 25(OH)D was measured using a 25OH total vitamin D Liaison^®®^ (DiaSorin, Saluggia, Italy) chemiluminescent immunoassay (CLIA) for quantifying 25(OH)D and had inter- and intra-assay coefficients of variation (CV) of 10.6% and 5.4%, respectively, with a lower detection limit (LOD) of <4 ng/mL or 10 nmol/L), as mentioned in previous studies [17,18]. It is worth noting that the CBCD is a participating laboratory of the DEQAS (Vitamin D External Quality Assessment Scheme). Serum Ca, albumin, Pi, and ALP were measured by routine laboratory analysis (Konelab, Vintaa, Finland). This biochemical analyzer was regularly calibrated before the analysis of all serum samples using quality control samples provided by the manufacturers (Thermo Fisher Scientific, Espoo, Finland). All biochemical analyses were performed at the CBCD at King Saud University, Riyadh, Saudi Arabia.

### 2.3. Definition of Biochemical Osteomalacia

For the purpose of this study, biochemical osteomalacia was defined as a combination of any two of the four serum markers of impaired mineralization [8], namely low 25 hydroxyvitamin D (25OHD < 30 nmol/L), high ALP (age- and sex-specific adjusted reference ranges obtained from the CALIPER study [19]), and either low calcium (<2.1 mmol/L) and/or low Pi (age- and sex-adjusted reference ranges) [20], as provided in Appendix A.

### 2.4. Data Analysis

Data were analyzed using SPSS version 21.0. BMI values were transformed into BMI Z-scores using reference means and standard deviations for Saudi school-age children and adolescents [21]. Categorical variables were presented as N (%). Normally distributed quantitative variables were presented as mean ± SD and, where data were not normally distributed, the median (interquartile range, IQR) was reported. For the purpose of this study, an absolute skewness value ≤2 or an absolute kurtosis (excess) ≤4 was used as a reference to determine normality [22,23]. Serum ALP and 25(OH)D were considered non-normal variables, and the normality of other parameters assessed is presented in Appendix A. An independent T-test and the Mann–Whitney U-test were used to determine statistical differences between groups for normal and non-normal quantitative variables, respectively. Logistic regression was used to determine the odds of biochemical osteomalacia for select risk factors. A *p*-value < 0.05 was considered significant.

## 3. Results

A total of 2938 adolescents participated in this study, divided into 1697 girls and 1241 boys. The mean age of study participants was 14.9 ± 1.7 years. Table 1 shows the descriptive statistics of anthropometric and biochemical parameters in boys and girls. Unsurprisingly, boys were slightly but significantly older and taller, with a higher BMI than girls. Furthermore, serum Ca, Pi, ALP, and 25(OH)D were significantly lower in girls than in boys (all *p*-values < 0.001).

Figure 1 shows the prevalence of the different components of osteomalacia assessed in the present study. The overall prevalence of biochemical osteomalacia was 10% (girls 14.7% vs. boys 3.6%, *p* < 0.001). The prevalence of vitamin D deficiency, hypocalcemia or hypophosphatemia, and high ALP were 43.9%, 19.2%, and 3.6%, respectively. More than half of the participants (52.2%) had at least one deranged component of biochemical osteomalacia. When stratified according to sex, girls had a significantly higher prevalence of all components of biochemical osteomalacia than boys (*p* < 0.001) (Figure 1). Only four participants (1.4%), all girls, had all four components of biochemical osteomalacia (not shown in the figure).

Table 2 shows the differences in parameters among girls and boys with or without biochemical osteomalacia. In girls, those with biochemical osteomalacia were significantly heavier (*p* = 0.004) with a higher BMI (*p* = 0.02), BMI z-score (*p* = 0.009), and ALP (*p* = 0.05). In boys, those with biochemical osteomalacia did not significantly differ from those without, both in anthropometry and ALP. Lastly, all participants with biochemical osteomalacia had significantly lower Ca, Pi, and 25(OH)D (all *p*-values < 0.001). It is worth noting that none of the participants were vitamin D sufficient (>50 nmol/L).

Table 3 shows the increased odds of components of biochemical osteomalacia in girls compared to boys, adjusted for age and BMI. Girls were 2.2 times more likely than boys to have low serum Ca and/or Pi (95% confidence interval [95% CI 1.7–2.7]; *p* < 0.001) and 3.3 times more likely to be vitamin D deficient (25(OH)D < 30nmol/L) than boys [odds ratio, OR (95% CI 2.6–3.6); *p* < 0.001). The odds of having elevated ALP were also higher in girls as compared to boys, with an OR of 4.2 (95% CI 2.4–7.3; *p* < 0.001). Similarly, girls were more likely to have at least one component of biochemical osteomalacia as compared to boys, with OR of 4.0 (95% CI 3.3–4.7; *p* < 0.001). Lastly, the odds of biochemical osteomalacia were 4.6 times higher in girls compared to boys (95% CI 3.3–6.4; *p* < 0.001).

## 4. Discussion

In the present cross-sectional study, biochemical osteomalacia, which was defined as having at least two markers of impaired mineralization, was observed to have an overall prevalence of 10% among Arab adolescents and was 4.6 times more common in girls (14.7%) than boys (3.6%). The prevalence of 14.7% in adolescent girls obtained in the present study is much higher than the preliminary prevalence obtained by Sulimani and colleagues in 2016 [14]. In the previous study involving 2000 adolescent Saudi girls, 1548 of whom had been analyzed for bone markers, including circulating serum 25(OH)D, the prevalence of osteomalacia-typical biochemical changes (25(OH)D < 25 nmol/L + elevated ALP) with clinical signs and symptoms suggestive of osteomalacia was 2.1% (33 out of 1548) [14]. Although bone biopsies were not performed, it was evident that the cohort had variable degrees of vitamin D inadequacy (<50 nmol/L), with evidence of secondary hyperparathyroidism observed in approximately 10% of subjects [14]. Some of the reasons for the discrepancy in the prevalence of osteomalacia include the difference in 25(OH)D cut-offs used [<25 nmol/L vs. <30 nmol/L (present study)] and that the above study took into consideration clinical features, such as bone pain and muscle weakness, attributes that were not assessed in the present study. Nevertheless, population studies in the Middle East and North African (MENA) region, including European studies involving immigrant children of Arabian and African ethnicity, have consistently shown a high incidence of nutritional rickets and osteomalacia in this group, partly because of the strict cultural practices prevalent in the region that restricts individuals, women in particular, from ample sunlight exposure [24,25].

In comparison to other epidemiologic studies on nutritional rickets and osteomalacia conducted elsewhere, it is crucial to consider the operational definitions used, since other studies assessed their estimates using either clinical, radiological, biochemical, or combinations of these methods [26]. In the US, the Rochester Epidemiology Project Data tallied the incidence of nutritional rickets from 1970 to 2009 and observed a sharp rise from 0 per 100,000 in 1970 to 24.1 per 100,000 in the year 2000, based on radiographs consistent with rickets with no evidence of non-nutritional causes in followed-up participants [27]. Data in India revealed a substantially higher incidence of nutritional rickets at 2700 per 100,000, again based on radiographs of more than 16,000 children and adolescents <18 years [28]. On the other hand, in the UK, the incidence of osteomalacia is 0.48 per 100,000 children under 16 years, based on radiographic and biochemical data collected from 2015 to 2017 [29]. Only one study investigated the prevalence of osteomalacia on biochemical diagnosis alone in 189 highschools students from Pakistan [30]. The obtained prevalence was 27% [30], much higher than the obtained 10% in the present study.

While the recent global consensus provided guidance on the prevention and management of nutritional rickets and detailed its etiology and risk factors [31,32], there are still many open research questions, in particular regarding the true prevalence of these conditions in different populations and regions. More importantly, the prevalence of undiagnosed morbidity from evolving rickets and osteomalacia is only detectable biochemically [32]. In the present study, the overall prevalence of 25(OH)D < 30 nmol/L was 43.9%. For a sunny country, such as Saudi Arabia, this prevalence is very high and indicates widespread sun avoidance and cultural full-body clothing as causative factors, but also a lack of food fortification. The prevalence of vitamin D deficiency among participants of the current study would be 100% if a higher cut-off for vitamin D deficiency (<50 nmol/L) was used [4,5]. The effect of vitamin D insufficiency on bone health in isolation is not known. Only following prolonged deprivation of calcium and vitamin D do typical biochemical markers of osteomalacia, such as ALP and PTH, start to rise in the bloodstream. In this study, high vitamin D deficiency prevalence in combination with these biochemical hypomineralization markers revealed, as expected, a much lower prevalence of biochemical osteomalacia. This reinforces the premise that although 25(OH)D is the current gold standard marker for vitamin D status and is useful in recognizing at-risk groups, it remains unsuitable as a standalone marker for the diagnosis of nutritional rickets or osteomalacia. Hypocalcemia, hypophosphatemia, high ALP, and secondary hyperparathyroidism are more specific biochemical markers for hypomineralization (nutritional rickets and osteomalacia) [33].

An interesting finding in the study is that boys with components of biochemical osteomalacia were significantly taller than their counterparts without biochemical osteomalacia. This may well correspond to some of the boys still undergoing the pubertal growth spurt when bone growth outpaces bone mineralization, which may potentially lead to a relative reduction in bone mineral content [34].

The main limitation of the study is that the prevalence of biochemical osteomalacia as defined here is suggestive at best of the histological presence of osteomalacia since gold standard transiliac bone biopsies were not performed to confirm this diagnosis. However, a study including bone biopsies in healthy populations would prove that ethical challenges necessitate reliance on biochemical markers. Circulating PTH was also not assessed since pre-analytic conditions were not satisfied. The majority of the participants’ blood samples were collected pre-pandemic before the study was halted due to lockdowns and was resumed only a year after rendering the samples non-viable for PTH analysis. Intact PTH, whether assessed using serum or plasma, is viable only for 8 days after separation [33]. It is worth noting that secondary hyperparathyroidism in children usually sets in when 25(OH)D levels drop below 34 nmol/L [35], thus approaching the deficiency range set by the global consensus group [31].

Dietary information was also not included in the present study given the inherent flaws of food frequency questionnaires and recall bias. The inclusion of dietary calcium and other related information, together with the analysis of other micronutrients for concomitant deficiencies, may provide a more accurate clinical burden of osteomalacia in this at-risk population. However, since the focus of the study was more on the biochemical aspect of osteomalacia, only biomarkers of impaired mineralization were studied. Recent investigations on the dietary intake of children and adolescents in Saudi Arabia reported that almost a third of adolescents frequently (≥2 times per week) consume fast food [36] and children under 12 years have an above-average intake of cholesterol and saturated fat [37], reflecting high consumption of fast food. Furthermore, local dairy products in Saudi Arabia which are considered good sources of vitamin D and calcium for the younger population contain substantially lower amounts of these micronutrients when compared to fortified products imported from the United States [38]. For instance, fresh milk purchased in Saudi markets contains 0–400 IU/L of vitamin D, as opposed to 400 IU/quarter in US markets, while orange juice contains no vitamin D content at all in Saudi markets, as compared to 400 IU/quarter of orange juice in the US [38].

Given the scarcity of regional evidence on osteomalacia prevalence among Arab adolescents and the lack of studies about osteomalacia among adolescent boys, in particular, the study nonetheless fills a significant gap in the current literature on acquired bone disease among understudied populations. Furthermore, given the high prevalence of rickets and accumulating evidence on the extremely high prevalence of vitamin D deficiency as compared to other regions in the world, Saudi Arabia and its homogenous adolescent population is an optimal place and population to assess the true burden of calcipanic bone disease. This high prevalence also calls for political action to consider food fortification with vitamin D which is cheap and effective [39,40], in particular since vitamin D supplementation programs alone have varying degrees of efficacy [41].

## 5. Conclusions

In summary, the prevalence of biochemical osteomalacia among Arab adolescents is relatively high compared to previous estimates within the population, particularly in girls. This study highlights gender inequalities since female teenagers are significantly more affected by the consequences of vitamin D deficiency than male teenagers. Not just supplementation, but also food fortification with vitamin D should be seriously considered to reach sufficient vitamin D levels in the population. Further studies should take into consideration accurate dietary intake assessments and other micronutrient deficiencies which, in combination with known biochemical markers of impaired mineralization, may provide a better picture of the true extent of osteomalacia burden within the Arabic adolescent population. It will also be interesting to perform Tanner staging in future investigations to assess the association of pubertal stage and osteomalacia risk in this population.

## Figures and Tables

**Figure 1 nutrients-14-05354-f001:**
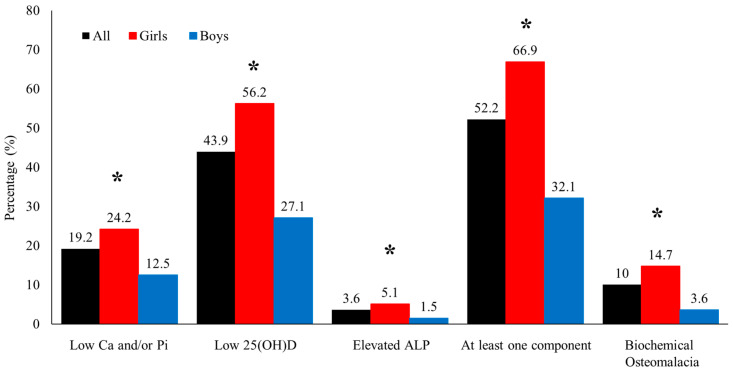
The prevalence of components of biochemical osteomalacia in all participants (black) and the significantly higher prevalence of each biochemical osteomalacia component in girls (red) than boys (blue). * denotes significance at *p* < 0.05.

**Table 1 nutrients-14-05354-t001:** Anthropometric and biochemical characteristics of study subjects.

Parameters	All	Girls	Boys	*p*-Value
N	2938	1697	1241	
Age (years)	14.9 ± 1.7	14.8 ± 1.8	15.1 ± 1.6	<0.001
Height (cm)	158.9 ± 10.3	156.4 ± 9.5	162.6 ± 10.3	<0.001
Weight (kg)	60.4 ± 16.8	57.6 ± 15.2	65.8 ± 33.6	<0.001
BMI (kg/m^2^)	23.8 ± 5.7	23.5 ± 5.7	24.2 ± 5.7	0.004
BMI Z-score	0.6 ± 1.2	0.6 ± 1.2	0.7 ± 1.2	0.06
Ca (mmol/L)	2.5 ± 0.4	2.5 ± 0.4	2.6 ± 0.4	<0.001
Pi (mmol/L)	1.5 ± 0.4	1.4 ± 0.3	1.6 ± 0.5	<0.001
ALP (U/L)	63.6 (42.5–94)	58.2 (39.5–88.9)	70.6 (48.0–100.6)	<0.001
25(OH)D (nmol/L)	30.8 (22.5–42.2)	26.9 (20.5–38.4)	35.8 (28.1–46.4)	<0.001

Note: Data are presented as mean ± standard deviation and median (IQR) for gaussian and non-gaussian continuous variables. The difference between the sexes is significant for *p* < 0.05. ALP, alkaline phosphatase; BMI, body mass index; Ca, calcium; Pi, inorganic phosphorous; 25(OH)D, 25(hydroxy) vitamin D.

**Table 2 nutrients-14-05354-t002:** Differences in study parameters according to biochemical osteomalacia status within gender.

Parameters	Girls			Boys		
	Controls	Cases	*p*-Value	Controls	Cases	*p*-Value
N	1447	250		1196	45	
Age (years)	14.7 ± 1.8	14.7 ± 1.7	0.90	15.1 ± 1.6	14.9 ± 1.5	0.59
Height (cm)	156.1 ± 9.5	157.1 ± 9.5	0.13	162.3 ± 10.5	163.7 ± 8.6	0.39
Weight (kg)	56.8 ± 15.2	59.9 ± 14.8	0.004	64.0 ± 18.1	67.7 ± 23.0	0.20
BMI (kg/m^2^)	23.3 ± 5.7	24.2 ± 5.9	0.02	24.1 ± 5.6	25.2 ± 8.3	0.22
BMI Z-score	0.6 ± 1.1	0.8 ± 1.1	0.009	0.7 ± 1.3	0.8 ± 1.2	0.53
Ca (mmol/L)	2.5 ± 0.3	2.3 ± 0.4	<0.001	2.6 ± 0.3	2.2 ± 0.5	<0.001
Pi (mmol/L)	1.5 ± 0.3	1.1 ± 0.3	<0.001	1.6 ± 0.5	1.2 ± 0.7	<0.001
ALP (U/L)	56.4 (38.7–84.4)	60.4 (40.7–96.4)	0.05	70.1 (47.6–98.3)	67.3 (55.5–110.6)	0.34
25(OH)D (nmol/L)	29.7 (21.4–41.4)	21.9 (18.0–25.9)	<0.001	37.0 (29.5–47.6)	21.3 (17.4–25.2)	<0.001

**Note**: Data are presented as mean ± standard deviation and median (IQR) for Gaussian and non-Gaussian continuous variables. *p* < 0.05 is considered significant. ALP, alkaline phosphatase; BMI, body mass index; Ca, calcium; Pi, inorganic phosphorous; 25(OH)D, 25(hydroxy) vitamin D.

**Table 3 nutrients-14-05354-t003:** Risk of biochemical osteomalacia in girls compared to boys.

Components	Crude OR (95%CI)	*p*-Value	Adjusted OR (95%CI) *	*p*-Value
Low Ca and/or Pi	2.1 (1.7–2.6)	<0.001	2.2 (1.7–2.7)	<0.001
Low 25(OH)D	3.1 (2.7–3.7)	<0.001	3.3 (2.8–3.9)	<0.001
Elevated ALP	3.3 (2.0–5.5)	<0.001	4.2 (2.4–7.3)	<0.001
At least one component	3.7 (3.2–4.4)	<0.001	4.0 (3.3–4.7)	<0.001
Biochemical osteomalacia	4.5 (3.3–6.3)	<0.001	4.6 (3.3–6.4)	<0.001

Note: * Adjusted for age and BMI; odds ratio, OR (95% confidence interval, CI) has been calculated as the probability of having the component in girls compared to boys. *p* < 0.05 was considered statistically significant. ALP, alkaline phosphatase; Ca, calcium; Pi, inorganic phosphorous; 25(OH)D, 25(hydroxy) vitamin D.

## Data Availability

Data are contained within the article.

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
