# Peer review of "Establishing the Prevalence of Osteomalacia in Arab Adolescents Using Biochemical Markers of Bone Health"

_nutrients, 2022, doi:10.3390/nu14245354_

Round 1
Reviewer 1 Report
Dear authors,
I really enjoyed reading your article about the importance of vitamin D supplementation and osteomalacia in adolescents. I have some recommendations and comments.
Introduction : are there other studies conducted on the pediatric population in Saudi Arabia regarding the prevalence of vitamin D deficiency and rickets? If there are, please present it.
Methods: adolescents with weight hypotrophy or obesity were excluded?
Discussion: when you refer to eating habits, maybe it would be good to detail a little more what the usual meal of a teenager in your country consists of.
- Also, what is the calcium and vitamin D content of the foods most frequently consumed in Arab countries?
-I think the long-term risks and effects of calcium and vitamin D deficiency in the pediatric population should be briefly presented.
-more comparisons with other studies carried out both in Arab countries and in other geographical areas should be discussed.
Author Response
Dear authors,
I really enjoyed reading your article about the importance of vitamin D supplementation and osteomalacia in adolescents. I have some recommendations and comments.
- Introduction: are there other studies conducted on the pediatric population in Saudi Arabia regarding the prevalence of vitamin D deficiency and rickets? If there are, please present it.
Response: We thank the reviewer for this comment. We expanded the introduction to provide additional studies, mostly retrospective, done in different KSA regions.
Page: 2 Line: 74-86
- Methods: adolescents with weight hypotrophy or obesity were excluded?
Response: These were not part of the exclusion criteria. We wanted apparently healthy children with no acute medical conditions independent of weight, so the findings can apply to the wider Saudi paediatric and adolescent population.
- Discussion: when you refer to eating habits, maybe it would be good to detail a little more what the usual meal of a teenager in your country consists of.
Response: We included additional statements in the revised discussion stating the high fast-food consumption of children and adolescents in Saudi to describe the typical diet of this population.
Page: 8 Line: 293-296
- Also, what is the calcium and vitamin D content of the foods most frequently consumed in Arab countries?
Response: Most processed dairy products purchased in Saudi stores are fortified with vitamin D, but this is less compared to US markets. This has been added in the revised discussion.
Page: 8 Line: 296-301
- I think the long-term risks and effects of calcium and vitamin D deficiency in the pediatric population should be briefly presented.
Response: This point is very well taken. We have added the long-term risks and effects of these micronutrient deficiencies in introduction.
Page: 2 Line: 67-69
- More comparisons with other studies carried out both in Arab countries and in other geographical areas should be discussed.
Response: The discussion was substantially expanded taking into consideration the reviewer’s valuable input on the inclusion of other similar studies performed elsewhere.
Page: 7 Line: 241-253
Reviewer 2 Report
This article titled “Establishing the Prevalence of Osteomalacia in Arab Adolescents Using Biochemical Markers of Bone Health”, is a cross-sectional study intended to determine osteomalacia presence among Arab adolescents using four (serum 25(OH)D, ALP, Ca and Pi) serum markers of mineralization.
It is very interesting, but some concerns were raised.
M&M
- there is a gap in inclusion/exclusion criteria
- line 97 - how explain how the nurse was trained? or calibrated? Just one nurse evaluated the patients?
- How was the researcher responsible to collect the data? Detail better this part.
- Line 120: “2.3 Definition of Biochemical Osteomalacia” - Where is the reference to consider this definition?
Results
- Excellent n (2938) was included
- lines 138-144: “Boys were slightly but significantly older and 141 taller, with higher BMI compared to girls.” Not relevant data, it was expected. Correct?
- Was observed any impact of the puberty period?
- Table 2: double-check the statistic for Pi and Ca. The numbers were close.
- Please recheck the statistical analysis for all data; show the normality and the raw data in a table within the article.
Discussion: needs to be improved; practically, the authors included just 2 paragraphs of discussion; 2 for limitations and 1 for the conclusion
- suggestion: include info from other countries to make a comparison
Author Response
This article titled “Establishing the Prevalence of Osteomalacia in Arab Adolescents Using Biochemical Markers of Bone Health”, is a cross-sectional study intended to determine osteomalacia presence among Arab adolescents using four (serum 25(OH)D, ALP, Ca and Pi) serum markers of mineralization. It is very interesting, but some concerns were raised.
M&M
- There is a gap in inclusion/exclusion criteria.
Response: This has been revised for clarity.
Page: 2 Line: 96-100
- Line 97 - how explain how the nurse was trained? or calibrated? Just one nurse evaluated the patients?
Response: We thank the reviewer for pointing this out. Each school has been assigned a research nurse to conduct anthropometrics and blood extraction.
Page: 3 Line: 110-113
- How was the researcher responsible to collect the data? Detail better this part.
Response: Data and sample collection were done mostly by research nurses assigned in each school as now described in subsection 2.1.
- Line 120: “2.3 Definition of Biochemical Osteomalacia” - Where is the reference to consider this definition?
Response: We thank the reviewer for raising this question. The operational definition was based on the markers of impaired mineralization proposed by Uday and Högler (2020) (ref 9). This has now been included in the revised subsection 2.3.
- Results: Excellent n (2938) was included. Lines 138-144: “Boys were slightly but significantly older and 141 taller, with higher BMI compared to girls.” Not relevant data, it was expected. Correct?
Response: We agree with the reviewer. We revised the statement accordingly.
Page: 4 Line: 158-160
- Was observed any impact of the puberty period?
Response: We thank the reviewer for this inquiry. It was assumed that all participants are at varying levels of pubertal growth, hence this was not considered. It will be interesting to perform Tanner staging in future investigations to assess the association of pubertal stage and osteomalacia risk in this population. We included this in the revised discussion.
Page: 9 Line: 319-320
- Table 2: double-check the statistic for Pi and Ca. The numbers were close.
Response: We double-checked the data for Pi and Ca and we confirm its accuracy. These markers have a very narrow range and usually appears close, especially if the data is presented with only one decimal point.
- Please recheck the statistical analysis for all data; show the normality and the raw data in a table within the article.
Response: Assumption of normality is now provided as supplementary table S2. Details have been provided in the revised data analysis (subsection 2.4).
Page: 3 Line: 149-151
- Discussion: needs to be improved; practically, the authors included just 2 paragraphs of discussion; 2 for limitations and 1 for the conclusion. Suggestion: include info from other countries to make a comparison.
Response: The revised discussion was expanded to include several data from other countries as rightfully suggested by the reviewer.
Page: 7 Line: 241-253; Page: 8 Line: 296-301
Round 2
Reviewer 1 Report
Dear authors,
I consider the manuscript is improved and ready for publication.
Reviewer 2 Report
Thank you so much for all clarifications and adjustments made.
I considered enough.
Congratulations.